# Examining the Effects of Hyperbaric Oxygen Therapy on the Cardiovascular System and Oxidative Stress in Insulin-Treated and Non-Treated Diabetic Rats

**DOI:** 10.3390/ani13182847

**Published:** 2023-09-07

**Authors:** Petar Ristic, Maja Savic, Sergey Bolevich, Stefani Bolevich, Alexandra Orlova, Anastasiya Mikhaleva, Anna Kartashova, Koka Yavlieva, Tamara Nikolic Turnic, Bozidar Pindovic, Katarina Djordjevic, Ivan Srejovic, Vladimir Zivkovic, Vladimir Jakovljevic

**Affiliations:** 1Clinic of Endocrinology, Military Medical Academy, 11000 Belgrade, Serbia; petris011@yahoo.com; 2Department of Pharmacy, Faculty of Medical Sciences, University of Kragujevac, 34000 Kragujevac, Serbia; tnikolict@gmail.com (T.N.T.); pindovic.bozidar@gmail.com (B.P.); kacka96kg@gmail.com (K.D.); 3Department of Human Pathology, I.M. Sechenov First Moscow State Medical University, 119991 Moscow, Russia; bolevich2011@yandex.ru (S.B.); alistra555@mail.ru (S.B.); orlovaas@yandex.ru (A.O.); stasy29051994@mail.ru (A.M.); kartashova.a@liceum1535.ru (A.K.); yavlieva-koka@mail.ru (K.Y.); drvladakgbg@yahoo.com (V.J.); 4Department of Pathophysiology, I.M. Sechenov First Moscow State Medical University, 119991 Moscow, Russia; 5N.A. Semashko Public Health and Healthcare Department, F.F. Erismann Institute of Public Health, I.M. Sechenov First Moscow State Medical University, 119991 Moscow, Russia; 6Department of Physiology, Faculty of Medical Sciences, University of Kragujevac, 34000 Kragujevac, Serbia; ivan_srejovic@hotmail.com (I.S.); vladimirziv@gmail.com (V.Z.); 7Departmennt of Pharmacology, Institute of Biodesign and Complex System Modeling, I.M. Sechenov First Moscow State Medical University, 119991 Moscow, Russia

**Keywords:** hyperbaric oxygen therapy, neutral protamine hagedorn (NPH) insulin, streptozotocin, type 1 diabetes (T1D)

## Abstract

**Simple Summary:**

This study aimed to examine the effects of hyperbaric oxygen therapy (HBOT) on the cardiovascular system and oxidative stress in streptozotocin-induced diabetic rats, with and without insulin treatment. Diabetes was induced in Wistar albino rats using streptozotocin. The rats were divided into four groups: DM group (diabetic rats), DM+HBOT group (diabetic rats exposed to HBOT for 1 h daily, five days a week, at 2.8 atmosphere absolute (ATA) with 100% oxygen for two weeks), DM+INS group (diabetic rats treated with neutral protamine hagedorn (NPH) insulin at a dosage of 3–5 U/day), and DM+HBOT+INS group (diabetic rats treated with both NPH insulin and HBOT for two weeks). Parameters evaluated included glycemic control, oxidative stress markers, and cardiac function. In conclusion, NPH insulin treatment reduced hyperglycemia and improved cardiac function in diabetic rats. The combination of insulin and HBOT exhibited a significant decrease in pro-oxidative markers. These findings contribute to our understanding of the potential benefits of HBOT and insulin therapy in managing cardiovascular complications and oxidative stress in diabetes, providing valuable insights for potential applications in clinical settings.

**Abstract:**

Background: This study explored the effects of hyperbaric oxygen therapy (HBOT) on the cardiovascular system and oxidative stress in streptozotocin-induced diabetic rats. Wistar albino rats were divided into four groups: DM group (diabetic rats), DM+HBOT group (diabetic rats exposed to HBOT for 1 h daily, five days a week, at 2.8 atmosphere absolute (ATA) with 100% oxygen for two weeks), DM+INS group (diabetic rats treated with neutral protamine hagedorn (NPH) insulin at a dosage of 3–5 U/day), and DM+HBOT+INS group (diabetic rats treated with both NPH insulin and HBOT for two weeks). Methods: Evaluations included glycemic control, oxidative stress parameters, and cardiac function measurements. Results: NPH insulin treatment reduced blood glucose levels, although normoglycemia was not achieved. The DM+HBOT+INS group demonstrated the lowest pro-oxidative marker levels. NPH insulin treatment improved cardiac function, and combination therapy effectively restored cardiac function in diabetic animals. Conclusions: NPH insulin treatment reduced hyperglycemia and improved cardiac function in diabetic rats. The combined approach of NPH insulin and HBOT resulted in decreased pro-oxidative markers. These findings provide valuable insights for managing cardiovascular complications and oxidative stress in diabetes.

## 1. Introduction

Diabetes is a chronic metabolic disorder characterized by impaired glucose homeostasis, oxidative stress, and an increased risk of cardiovascular complications. Since its difficulties continue to affect the human population, several animal models are constantly being created to explore the pathogenic process and therapeutic approaches. In experiments with rats and other animals, the drug streptozotocin (STZ) is frequently used to cause diabetes. It was initially identified as having a diabetogenic effect in 1963, and since then, it has been employed in several research studies in a variety of combinations to induce diabetes. STZ has a short half-life, due to rapid metabolism in the liver and elimination by renal excretion. Once STZ is eliminated from the body, further functional impairment of the liver or the kidney may be attributed to the effects of diabetic hyperglycemia. This is the basis for studying the mechanisms of STZ diabetic complications in these organs as well as other organs such as the heart in our case [1]. 

The management of diabetes includes lifestyle modifications, pharmacological interventions, and emerging therapies such as hyperbaric oxygen therapy (HBOT). HBOT involves exposing patients to high levels of oxygen in a pressurized chamber, leading to various physiological effects.

HBOT involves exposing the patient to a higher barometric pressure than normal (1 atmosphere absolute or 1 ATA) while inhaling 100% O_2_. HBOT increases the amount of dissolved oxygen in the blood, which improves a range of clinical situations such as hypoxia, acute carbon monoxide intoxication, air embolism, and diabetic lower limb lesions [2]. HBOT has emerged as a therapeutic intervention for various medical conditions. The elevated production of reactive oxygen species (ROS) during HBOT is considered beneficial as it stimulates the synthesis of crucial growth factors like vascular endothelial growth factor (VEGF), placental growth factor (PGF), and angiopoietin (Ang) 1 and 2 and facilitates the recruitment of stem cells from the bone marrow, thereby promoting neovascularization [3,4]. So it appears that HBO exposure is a great model system for research into oxidative stress.

The term “ischemia-reperfusion injury” (IR injury) refers to the worsening of cellular damage and dysfunction after reintroducing blood flow to previously ischemic tissues. ROS have been identified as the primary mediators of this phenomenon. During IR injury, the blood–endothelial-cell interface exhibits enhanced microcirculatory neutrophil adhesion, which causes tissue necrosis and initiates a feedback loop that culminates in more ROS production and injury [5]. However, preserving ischemic tissues requires restoring blood flow as soon as feasible. Given that IR is a paradoxical process in which reperfusion causes more tissue damage, which is dependent on ROS, it was assumed that a hyperoxic tissue environment given by HBOT following IR injury would enhance ROS production and aggravate the extent of tissue necrosis. Unexpectedly, HBOT has been demonstrated to upregulate antioxidant gene expression in human endothelial cells, perhaps protecting against the oxidative damage seen in IR injury [6,7,8]. As a result, HBOT appears to benefit ischemic tissue by lowering ROS generation while enhancing ROS degradation. HBOT has also been demonstrated to increase the production of heme oxygenase-1 (HMOX1), a protein that defends cells against oxidative stress, in both human beings and cell culture models. This result supports HBOT’s potential efficacy as a preconditioning therapy for boosting tolerance to a variety of ailments, especially those with an oxidative stress component. Such preconditioning may be carried out before a procedure that entails the possibility of an ischemia/reperfusion occurrence [9].

The purpose of this study was to determine the effects of HBOT on glucose homeostasis, oxidative stress parameters, and cardiac function in experimentally induced type 1 diabetes (T1D) and neutral protamine hagedorn (NPH)-insulin-treated and non-treated rats. The choice of using NPH insulin treatment was based on its widespread use in clinical practice for managing diabetes, and understanding how HBOT affects cardiac function can provide valuable information for developing targeted therapies to mitigate the adverse cardiovascular effects of diabetes.

## 2. Materials and Methods

### 2.1. Ethical Statement

The University of Kragujevac in Serbia’s Faculty of Medical Sciences conducted this inquiry (approval number 01-10540/1). The study procedure was carried out in compliance with the European Council Directive (86/609/EEC) and the ethical guidelines established by the faculty’s ethical committee for laboratory animal welfare.

### 2.2. Animals and Experimental Design

Forty-eight Wistar albino rats (8-week-old males, body mass 200 ± 50 g) were included in the study. The rats were housed at a temperature of 22 ± 2 °C with 12 h of automatic illumination each day and were fed commercial rat food freely. Rats were randomly divided into 4 groups (n = 12): (1)DM group: diabetes induced by streptozotocin (STZ) injection;(2)DM+HBOT group: received both STZ injection and HBOT, exposed to 100% oxygen at 2.8 ATA (atmosphere absolute) for 1 h once daily, for 5 days (two weeks);(3)DM+INS group: NPH insulin 5 U/day;(4)DM+HBOT+INS group: received both NPH insulin and HBO exposure for 2 weeks.

### 2.3. Induction of Diabetes

The rats were subjected to 24 h starvation and given an intraperitoneal injection of streptozotocin (STZ, 60 mg/kg body weight, dissolved in 0.01 M sodium citrate buffer, pH 4.5) to induce diabetes [10]. After an overnight fast and a streptozotocin injection, diabetes was determined by monitoring the blood glucose level in the tail veins with a portable glucometer. Animals were used as wound models that had diabetes when their blood glucose levels were higher than 11.1 mmol/L.

### 2.4. Protocol of Human NPH Insulin Treatment

Glycemic level control of diabetic rats was performed with subcutaneous injections of exogenous human NPH insulin [11]. The objective of the insulin therapy was to keep the glycemia of these animals as close as possible to the normoglycemia (from 3.3–8.3 mmol/L) throughout the 24 h day. Initially, the administration chosen was 4 U/day of NPH insulin. Throughout the treatment, daily dose of insulin was adjusted on average every 3 days according to the glycemia of each animal (from 3 to 5 U/day).

### 2.5. HBO Treatment 

A specially constructed hyperbaric pressure chamber (HYB-C 300, Maribor, Slovenia) was used for the experiment. Rats were exposed to 100% oxygen once daily for 14 days for 60 min at a pressure of 2.5 ATM and a flow rate of 7 L/min ± 10%. To avoid the effects of diurnal rhythm variations, hyperbaric oxygenation always began at the same time [12]. 

### 2.6. Biochemical Assays

#### 2.6.1. TBARS Determination (Index of Lipid Peroxidation) 

Thiobarbituric acid-reactive compounds (TBARS) were detected at 530 nm by incubating coronary effluent with 1% thiobarbituric acid in 0.05 M NaOH for 15 min at 100 °C to determine the level of lipid peroxidation. The blank probe was made of distilled water. TCA (trichloro acetic acid) and 0.8 mL coronary effluent were combined to create the TBA extract, which was then centrifuged for 15 min at 6000 rpm. This process was previously explained by [13].

#### 2.6.2. Nitrite Determination 

Rapidly oxidizing nitric oxide produces stable metabolite nitrite/nitrate compounds which were determined spectrophotometrically at 543 nm using Griess’s reagent according to Green’s method. Sodium nitrite was used as the reference standard to calculate the nitrite levels [14].

#### 2.6.3. Hydrogen Peroxide Determination

Based on the oxidation of phenol red by hydrogen peroxide, a process catalyzed by horseradish peroxidase (HRPO) is used to measure hydrogen peroxide (H_2_O_2_). Following the addition of 10 µL of (1:20) HRPO, 200 µL of perfusate was precipitated with 800 mL of newly produced phenol red solution (made extempore). Krebs–Henseleit solution was used as blank. H_2_O_2_ concentration was determined at 610 nm [15].

#### 2.6.4. Superoxide Anion Radical Determination 

After coronary effluent samples were incubated with nitro blue tetrazolium in Tris buffer, the quantity of the superoxide anion radical (O_2_^−^) was determined and read at 530 nm. The blank was Krebs–Henseleit solution [16].

### 2.7. Ex Vivo Monitoring of Cardiac Function

Rats were anesthetized with intraperitoneal injection mixture of 10 mg/kg ketamine (Pfizer Pharmaceuticals, New York, NY, USA) and 100 mg/kg xylazine (Interchemie, Castenray, The Netherlands) [17] and slaughtered by decapitation a day after the two-week treatment procedure was finished. Then, the chest was opened using midline thoracotomy. The hearts were promptly removed, immersed in cold saline, and connected to a Langendorff apparatus cannula for continuous coronary perfusion and retrograde perfusion (coronary perfusion pressure (CPP) 70 cmH_2_O). Krebs–Henseleit buffer, which had a pH of 7.4, a temperature of 37 °C, and a balance of 95% O_2_ and 5% CO_2_, was used for retrograde perfusion. After inserting the sensor into the left ventricle, the heart rate (HR), the systolic left ventricular pressure (SLVP), the diastolic left ventricular pressure (DLVP), the maximum rate of pressure development in the left ventricle (dp/dt max), and the minimum rate of pressure development in the left ventricle (dp/dt min) were all continuously measured. The coronary flow was measured by flowmetry (CF). The hearts were stabilized 30 min after cardiac perfusion was begun. Following a stabilization period, each group’s hearts underwent 20 min of global ischemia (in which blood flow was interrupted), followed by 30 min of reperfusion. All cardiodynamic parameters and CF were monitored at 5 min intervals during the 30 min reperfusion phase [18].

### 2.8. Statistical Analysis

Statistical analysis was conducted using SPSS 22.0 statistical package. Data are presented as mean ± standard deviation (SD). We performed the one-way ANOVA and Bonferroni test for multiple comparisons of parameters of systemic redox status and cardiodynamic parameters in each point between four groups. Additionally, in our study, data generated from time-course measurements such as changes in glycemic level over time (before treatment and 2-weeks after treatment) were analyzed using paired *t*-test. The accepted statistical significance values were *p* < 0.05. For all oxidative stress markers and groups, we calculated the AUC using statistical methods in SPSS.

## 3. Results

### 3.1. Glycemia

NPH insulin treatment significantly reduced blood glucose levels after two-week treatment. NPH does not induce normoglycemia but significantly reduced hyperglycemia in treated groups (DM+INS and DM+INS+HBOT). Even HBOT alone led to a slight lowering of glycemia but without statistical significance (Figure 1). 

### 3.2. Biomarkers of Oxidative Stress

Concentrations of pro-oxidation markers (O_2_^−^, H_2_O_2_, TBARS, and NO_2_^−^) were determined from the coronary venous effluent of diabetic rats treated with NPH insulin, HBOT, or their combination and collected at the moment of stabilization (S), then in the first minute of reperfusion after ischemia (R), and then every 5 min during the thirty-minute reperfusion, and their values are shown graphically (Figure 2).

#### 3.2.1. Superoxide Anion Radical (O_2_^−^)

The superoxide anion radical level was the lowest in the DM+INS+HBOT group in almost all measurement points, both during stabilization and during reperfusion following ischemia. The highest jump in the superoxide anion radical level was recorded in the 1st and 30th minute of reperfusion in the DM group and the DM+INS group. Combined treatment with NPH insulin and HBOT significantly suppressed the production of superoxide anion radicals, especially in the first minutes of reperfusion (Figure 2a).

#### 3.2.2. Hydrogen Peroxide (H_2_O_2_)

The DM group had the highest hydrogen peroxide levels during all 30 min of reperfusion. Treatment with NPH insulin or HBOT led to a slight decrease in the value of hydrogen peroxide compared to the DM group, but without statistical significance, while their combination (DM+INS+HBOT group) significantly decreased the level of hydrogen peroxide in the coronary venous effluent of rats, during all 30 min of reperfusion (Figure 2b). A similar trend was recorded for TBARS and nitric oxide values (Figure 2c,d).

#### 3.2.3. Index of Lipid Peroxidation (Measured as TBARS)

The index of lipid peroxidation values was the highest in the DM group, and all interventions, treatment with NPH insulin, HBOT, or their combination, led to a significant decrease in values during reperfusion. By far, the lowest values were recorded in the DM+INS+HBOT group during the last 20 min of reperfusion (Figure 2c). 

#### 3.2.4. Nitric Oxide (NO)

As with the previous parameters of oxidative stress, the value of nitric oxide was the lowest in the DM+INS+HBOT group. NPH insulin alone did not significantly decrease nitric oxide production, while HBOT led to a significant decrease, as did NPH insulin treatment alone (Figure 2d).

#### 3.2.5. AUC for Markers of Oxidative Stress

For all markers in all groups, we calculated the area under the curve based on the estimated concentrations of oxidative stress markers (Figure 3). We observed for almost all markers good availability which was in the range above 0.7. The only exception was observed for the superoxide anion radical, which confirmed poor availability (AUC = 0.682) which is expected, since this free radical has a very short time of existing in the in vivo and ex vivo conditions. Our results of the AUC represent a good bioavailability of biomarkers of oxidative stress, and slightly lower results could be a result of interindividual variability of the rat population. However, the values of each oxidative stress marker in relation to the group ranged ± 0.5 AUC, so there was no statistical difference between the groups in the AUC.

### 3.3. Estimation of Cardiac Function 

Cardiodynamic parameters and coronary flow of diabetic rats treated with NPH insulin, HBOT, or their combination after 20 min of ischemia followed by thirty minutes of reperfusion are shown graphically (Figure 4).

#### 3.3.1. Minimum Rate of Left Ventricular Pressure Development (dp/dt min)

The value of dp/dt max was the lowest in the DM group in all measurement points, both during stabilization and during reperfusion following ischemia. The greatest leap in value dp/dt max was recorded in the 15th minute of reperfusion in the DM+INS+HBOT group in comparison with all other groups. Treatment with NPH insulin alone or in combination with HBOT induced an increase in the value of dp/dt max in almost all measurement points in comparison to the DM and DM+HBOT groups (Figure 4a).

#### 3.3.2. Minimum Rate of Left Ventricular Pressure Development (dp/dt min) 

The absolute values of dp/dt min of rats from the DM+INS+HBOT group were statistically significantly higher compared to all other groups at the point of stabilization as well as during reperfusion. Even the application of NPH insulin alone led to an increase in the absolute values of dp/dt min in comparison to the DM group, from the 10th to 25th minute of reperfusion (Figure 4b).

#### 3.3.3. Systolic Left Ventricular Pressure (SLVP) 

The values of SLVP in the group treated with NPH insulin alone and in combination with HBOT were statistically significantly higher compared to the values in the DM group at the point of stabilization as well as during reperfusion, except in the 30th minute of reperfusion, in which SLVP values were close in all groups. The application of HBOT alone did not significantly affect the values of SLVP in relation to the values recorded in the DM group (Figure 4c).

#### 3.3.4. Diastolic Left Ventricular Pressure (DLVP)

Combined therapy with NPH insulin and HBOT led to the most significant increase in DLVP values compared to all other groups, from the 10th to the 30th minute of reperfusion, while HBOT independently led to a significant increase in DLVP values in the last 10 min of reperfusion compared to the DM group (Figure 4d).

#### 3.3.5. Heart Rate (HR) 

NPH insulin therapy alone and in combination with HBOT led to an increase in HR values in the last 10 min of reperfusion compared to the DM group (Figure 4e).

#### 3.3.6. Coronary Flow (CF)

The DM group had the lowest CF during all 30 min of reperfusion as well as at stabilization. HBOT treatment led to a slight increase in the values of CF compared to the DM group but without statistical significance, while NPH insulin or combination (DM+INS+HBOT group) significantly increased CF during all 30 min of reperfusion (Figure 4f).

## 4. Discussion

Diabetes is distinguished by the development of particular microvascular problems [19]. Despite the fact that numerous studies have explored and contrasted the roles of the various components that contribute to diabetic vascular problems, an exact assessment of their respective contributions remains difficult [20]. Nevertheless, as several trials have shown, both microvascular and macrovascular consequences are mostly or partially dependent on hyperglycemia [21]. Hyperglycemia-induced vascular damage involves at least four major pathways: increased polyol activity, which causes sorbitol and fructose accumulation; increased formation of advanced glycation end products; activation of protein kinase C and nuclear factor kB; and increased hexosamine pathway flux. There are numerous reasons to believe that hyperglycemia causes all of these harmful metabolic processes through a single pathway: because the mitochondrial electron-transport chain produces too much superoxide. This theory proposed by Brownlee [22] appears to show that hyperglycemia-induced oxidative stress plays a significant role in the etiology of diabetic problems. The development of diabetes is significantly influenced by chronic inflammation. The presence of inflammation triggers the production of ROS by tissue or immune cells in a variety of peripheral tissues, including the liver, skeletal muscle, and adipose tissue. These ROS molecules bind with the insulin receptor (IR) and interfere with its downstream signaling pathways, preventing the body from responding to insulin levels in the proper manner [23]. It is well known that type 1 diabetes (T1D) may be studied in rats that have been made diabetic by the drug streptozotocin (STZ).

To complete the picture of the role of HBOT and glycemic control, in physiology and myocardial pathophysiology, using the Langendorff apparatus, through the function of the left ventricle, which is the morphologically and functionally more dominant heart cavity, myocardial function was monitored ex vivo. In addition, IR injury was induced on the Langendorff apparatus, and cardiac function was observed during the reperfusion period. In this way, HBOT was examined and studied, as a potential preconditioning agent. The augmentation of fibroblast replication, enhanced collagen production, and neovascularization of ischemic tissues are the most significant benefits of HBOT for IR damage. HBOT has been proven to raise arterial blood’s oxygen tension. Increasing the tissue-cellular diffusion gradient increases cellular oxygen supply, and hyperoxia enhances angiogenesis by increasing the creation of the necessary collagen matrix for angiogenesis [24].

For this purpose, oxidative stress parameters were measured in the coronary venous effluent of the isolated rat heart. TBARS, which are lipid peroxidation markers, have shown to be considerably elevated in this model, showing the existence of oxidative stress [25], and it is clearly seen that the combined treatment with NPH insulin and HBOT leads to a significantly reduced production of TBARS in the coronary venous effluent. As a byproduct of lipid peroxidation, MDA is sometimes seen as a sign of oxidative stress. Ilhan et al.’s findings [26] that HBOT delivered prior to ischemia improved renal I/R by lowering radical oxygen peroxidation of lipid membranes are consistent with our findings. However, there are data that contradict our findings. For instance, in [27], although the level of MDA did not reach a statistically significant level, it was greater in the groups getting HBOT than in the other groups. In this study, the rise in MDA levels was seen as a marker of oxidative damage that was supposed to arise from increased oxygen levels in the cells as a result of HBOT. A balance between the oxidant and antioxidant systems can be achieved by raising antioxidant levels in the same groups, which can reduce peroxidative damage. Also, it has been demonstrated that 30 min after HBOT with 30% oxygen pressure at 1.3 ATA in dogs, GPx activity in erythrocytes increased. The nuclear factor erythroid 2-related factor 2 (Nrf2), a redox-sensitive transcription factor, is thought to play a key role in the cellular defense against oxidative stress by upregulating the transcription of antioxidant stress proteins like GPx and phase II defense enzymes [28]. Additionally, in the liver tissues of STZ-induced diabetic rats, oxidative-stress-related changes in the mRNA and protein expression of antioxidant enzymes, such as copper-zinc superoxide dismutase (Cu-Zn SOD) and catalase (CAT), have been found [29]. These results imply that the pathological manifestations of diabetes offer a perfect setting for studying the processes behind oxidative stress. In the current study, we showed that a 2-week treatment with NPH insulin alone or in combination (even more pronounced) induces decreased production of pro-oxidation markers in the coronary venous effluent of rats, which was collected after 20 min of global ischemia (in which blood flow was interrupted), followed by 30 min of reperfusion. The acquired results are in line with those of a research study that examined how HBOT affected individuals with diabetes and demonstrated that HBOT did not have a pro-oxidative effect. The antioxidant enzyme system appears to have been highly recruited by this process as well, enhancing protection against oxidative damage [30].

Zamboni et al. [6,7,8] demonstrated that HBOT upregulates antioxidant gene expression in human endothelial cells, protecting ischemic tissue against oxidative damage by lowering ROS generation. Additionally, it has been demonstrated that seven daily HBOTs can reduce the expression of hypoxia inducible factor 1 alpha in the tissue of rat ischemic wounds, hence reducing apoptosis. Also, these data showed that after HBOT, fewer neutrophils were found at the wound site, indicating that inflammation had decreased. It is amazing that molecular chaperone expression is elevated in microvascular endothelial cells in response to HBOT. Although the precise role of these chaperones in endothelial cells is not yet known, our functional studies indicate that they may help shield cells from harmful oxidative stress. Pathway analysis identified the Nrf-2 oxidative stress response pathway as a crucial mechanism controlled by HBOT in terms of endothelial cell protection. More than 200 genes involved in protection and antioxidant activity are regulated by this pathway. Numerous Nrf-2 target genes can be made to express when exposed to HBOT, and it is probable that these genes are helping to protect endothelium cells from oxidative stress. Nrf-2 can control metallothioneins, which are among the best responders to HBOT. The heme oxygenase gene, HMOX1, is one fascinating Nrf-2-regulated gene that HBOT activates [9].

In terms of diabetes compensation, repeated HBOT throughout the year reduces insulin consumption, recovers residual insulin secretion, and suppresses the release of contra-insular hormones such as glucagon, somatotropic hormone, and hydrocortisone. It is more effective to administer three courses of HBOT to patients with insulin-dependent diabetes at intervals of four months rather than two courses separated by six months; however, in a three-course modality, the maximum beneficial effect of HBOT on the hormonal metabolic status is only achieved during the second course, and the third course merely reinforces the effect already attained [31]. The combined treatment of HBOT and NPH insulin demonstrated not only a beneficial effect on the coronary oxidation status but also positive effects on the cardiodynamics of the isolated rat heart. SLVP and dp/dt max are two cardiodynamic parameters that describe the systolic function, while the parameters dp/dt min and DLVP are related to the diastolic function of the heart; all four mentioned cardiodynamic parameters, as well as CF, were improved in the group of rats treated with the combined treatment (DM+INS+HBOT), which is in accordance with the results of similar studies [32,33]. It has been shown that HBOT induces a slow decrease in HR [32], but this was not recorded in our research.

## 5. Conclusions

After a two-week treatment, NPH insulin alone was unable to achieve normoglycemia but did effectively reduce extreme hyperglycemia in the diabetic rats. Notably, the DM+INS+HBOT group exhibited the most pronounced antioxidative benefits, followed by the HBOT group. Additionally, NPH insulin treatment significantly improved cardiac function, and the combination of NPH insulin and HBOT demonstrated effectiveness in restoring cardiac function in the diabetic animals.

## Figures and Tables

**Figure 1 animals-13-02847-f001:**
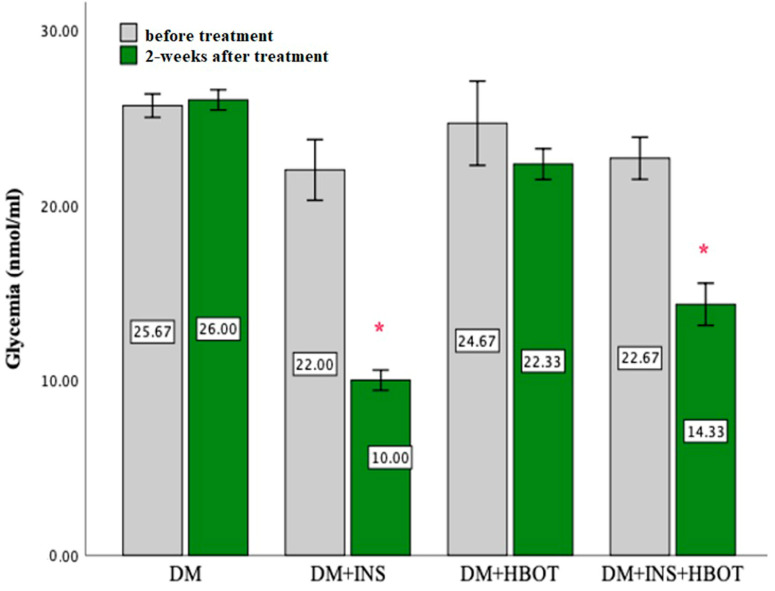
Effects of DM+INS, DM+HBOT, and DM+INS+HBOT treatment on levels of glycemia in rats before and after a 2-weeks treatment. Data are presented as means ± SD. The paired *t*-test for testing the differences in values of glycemia was used. * *p* < 0.05 within−group difference (DM, DM+INS, DM+HBOT, and DM+INS+HBOT) after 2-week treatment.

**Figure 2 animals-13-02847-f002:**
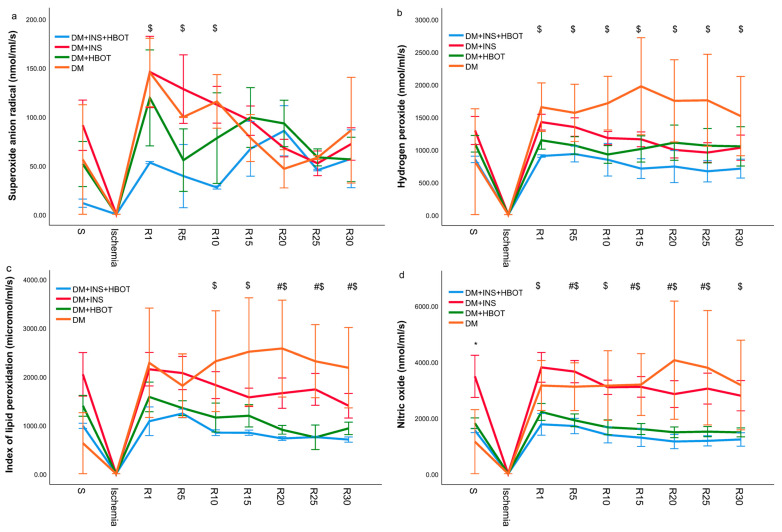
Effects of DM+INS, DM+HBOT, and DM+INS+HBOT treatment on oxidative stress. Comparison of the levels of (**a**) superoxide anion radical, (**b**) hydrogen peroxide, (**c**) index of lipid peroxidation (measured as TBARS), and (**d**) nitric oxide between experimental groups and DM (control) group. S—stabilization; R—every 5 min of reperfusion. Data are presented as means ± SD. The one-way ANOVA for testing the differences in biomarkers of oxidative stress was used. * *p* < 0.05 DM group vs. DM+INS; # *p* < 0.05 DM group vs. DM+HBOT; $ *p* < 0.05 DM group vs. DM+INS+HBOT.

**Figure 3 animals-13-02847-f003:**
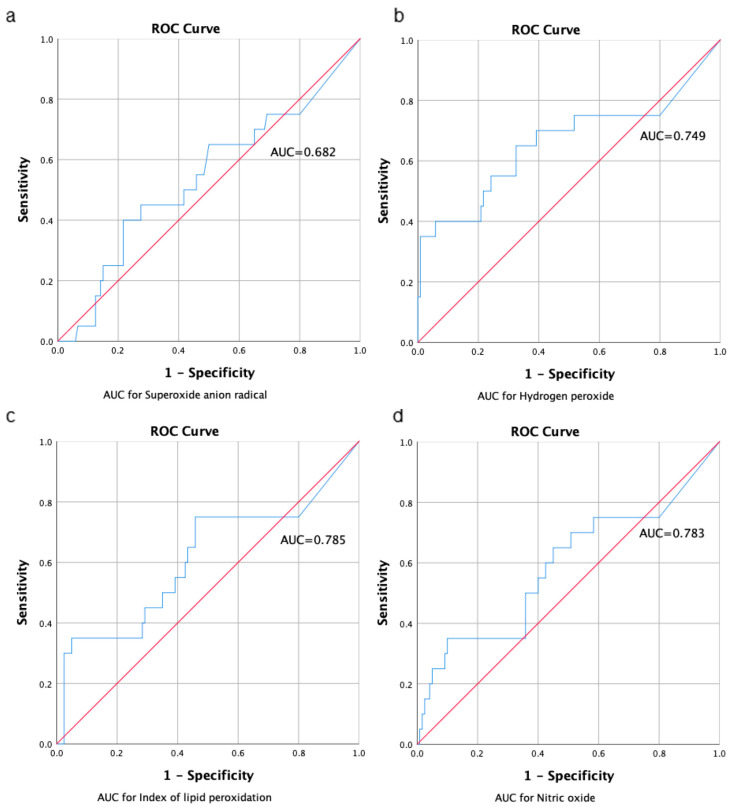
Area under curve for markers of oxidative stress calculated from ROC curve as follows: (**a**) superoxide anion radical, (**b**) hydrogen peroxide, (**c**) index of lipid peroxidation, and (**d**) nitric oxide. Levels of AUC higher than 0.7 present positive (good) bioavailability. All calculations are performed in SPSS program and are expressed in mg·h/L.

**Figure 4 animals-13-02847-f004:**
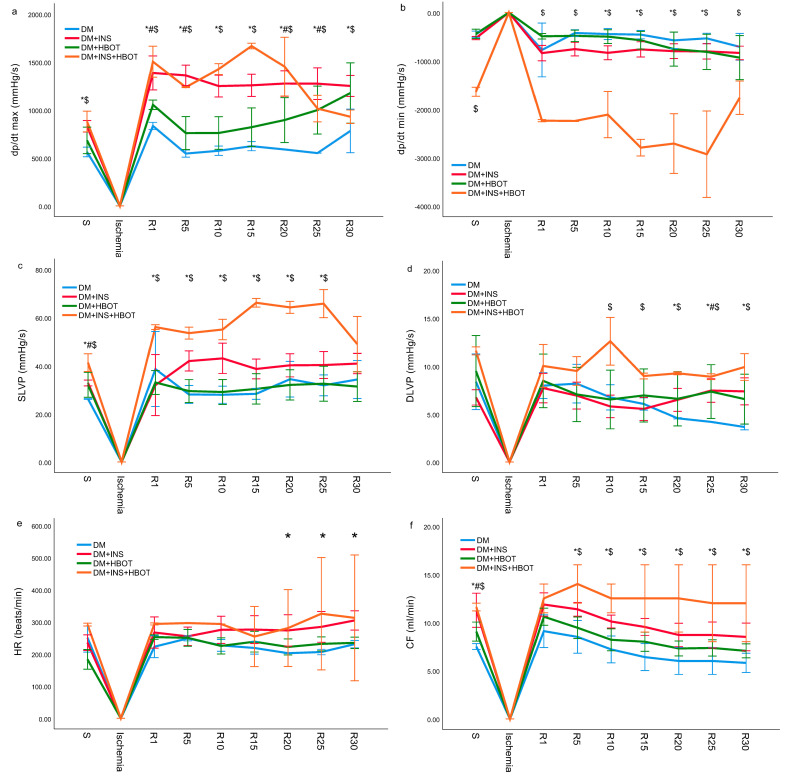
Effects of DM+INS, DM+HBOT, and DM+INS+HBOT treatment on cardiac function measured *ex vivo*. Comparison of (**a**) dp/dt max value, (**b**) dp/dt min value, (**c**) SLVP value, (**d**) DLVP value, (**e**) HR value, and (**f**) CF value, between experimental groups and control group. S—stabilization; R—every 5 min of reperfusion. Data are presented as means ± SD. The one−way ANOVA for testing the differences in biomarkers of cardiodynamic parameters was used. * *p* < 0.05 DM group vs. DM+INS; # *p* < 0.05 DM group vs. DM+HBOT; $ *p* < 0.05 DM group vs. DM+INS+HBOT.

## Data Availability

Not applicable.

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
