# Peer review of "Examining the Effects of Hyperbaric Oxygen Therapy on the Cardiovascular System and Oxidative Stress in Insulin-Treated and Non-Treated Diabetic Rats"

_animals, 2023, doi:10.3390/ani13182847_

Round 1

Reviewer 1 Report

The paper by Petar Ristc and cols, assesses the potential effects of HBOT on the cardiovascular system 2 and oxidative stress in insulin-treated and non-treated diabetic rats. This is a very interesting topic. The methods were remarkable, diabetes is an ever-changing challenge with devastating outcomes in the word.

The manuscript needs a minor revision and I have several considerations to make at the Authors:

Simple Summary and Abstract.   Define ATA and, NPH insulin.

Keywords. In alphabetical order.

In general: According to the ADA, the term diabetes mellitus is no longer used. Diabetes, type 2 diabetes (T2D) and type 1 diabetes (T1D) are now used. Please proofread the entire manuscript.

Introduction:

Line 68. HBOT was already previously defined

Line 69. O2 must be with subscript (in the entire manuscript)

Materials and methods:

Line 109. Rats were randomly divided into 4 groups. Replace by rats were randomly divided into 4 groups (n=12).

Line 1013. Define NPH Insulina

Line 132. Missing to indicate the units of concentration (0.05 NaOH)

.Line 146. "Krebs-Henseleit solution was",  It seems to be an incomplete sentence.

Results:

Fig 2 (line 268) and 3 (line 285). Eliminate  I - ischemia.

Discussion:

Line 291. TBARS was already previously defined.

Reviewer 2 Report

1. The discussion section of the manuscript should give more information about why cardiac parameters and oxidative stress markers were measured and the relationship between them.

2. At the simple summary section, authors indicate that insulin is used on 5 U/day , but at section 2.4, it was written "daily insulin dose adjusted according to glycemia of each animal (from 3 to 5 U/day)". Could you give more information about insulin dosage?

3. At the section 2.2, rats were divided into four groups (DM, DM+HBOT, DM+INS, DM+HBOT+ INS). Could you explain why there is not control group in experimental design?

4. Information should given about protocol of hyperbaric oxgen treatment in materials and methods section.

5. In the references section, all dates of literature must be bold (e.g. lit.14)

Reviewer 3 Report

The manuscript entitled „Examining the effects of HBOT on the cardiovascular system and oxidative stress in insulin-treated and non-treated diabetic rats” describes the effect of hyperbaric oxygen therapy on glucose concentration in diabetic rats. In addition, the study also assumed the demonstration or non-demonstration of the effect of this procedure on the effectiveness of insulin therapy. The study focused mainly on the cardiovascular system and oxidative stress. The manuscript is relatively interesting. However, it contains several errors/imperfections that should be corrected before its publication.

Major:

·       Why did the authors study only glucose concentration? In order to prove the correctness of the model used, they should take into account other factors such as: ip or oral ITT, IGT and insulin.

·       Please explain why the authors did not include a control group of healthy rats in their study. I mean not necessarily in the whole study, but to show the correctness of the research model (at the stage of induction of DM)

·       Please describe procedures in M&M section more precisely.

·       The authors did not specify what dose of ketamine and xylazine they used to kill the animals, so there is no certainty that it was not toxic and did not affect ex vivo heart function.

·       STATISTICS. The authors did not describe what tests were used for statistical analyzes in the M&M section. In addition, under figure 1 there is a caption that the changes were analyzed by the T-test. The wording is also surprising: "Independent T test for testing the differences in biomarkers of oxidative stress was used". Can the authors explain on what basis they qualified glucose as a marker of oxidative stress?

·       Statistics - figure 2 Why did the authors compare the 4 groups using the t test? The T test is a test to compare only 2 groups with each other.

·       The conclusion is not confirmed by the results because the authors did not study the changes compared to the group of healthy rats.

Minor:

·       Line: 145 author: According to the description, the authors added 10 l of HRPO. Was there a mistake here?

·       The authors are asked to change the units of glucose measurements, I understand that the values will not change, but the standard units for glucose are mmol/l and mg/dl.

·       I suggest that the authors calculate the AUC for markers of oxidative stress as well as in cardiac function measured ex vivo. This will allow for better visualization of the presented results.

Round 2

Reviewer 2 Report

The corrections are enough for publishing the manuscript.

Author Response

We appreciate the valuable suggestions provided to enhance our manuscript.

Reviewer 3 Report

The authors corrected and addressed most of the suggestions, but the authors did not include the AUC comment, explaining that they are not able to calculate these values in such a short time. 

I believe they should be included and it doesn't take much time to do so. However, if the Editor feels that this remark is redundant, I will not mind publishing the manuscript. However, I think it would greatly enrich its reception.

Author Response

Respected,

We sincerely appreciate your valuable feedback on our study. We have taken your input into careful consideration. Regarding the AUC analysis, we have successfully performed it for the parameters related to oxidative stress. However, we encountered limitations when attempting to conduct the AUC analysis for the cardiodinamics parameters. Since these parameters are functional in nature, they lack concentrations required for the AUC calculation. We have included this information in our methodology section to provide transparency on the scope of our analysis. We thank you once again for your insightful comments, which have contributed to enhancing the clarity and comprehensiveness of our research.